# Morphological Description and Classification of Wheat Kernels Based on Geometric Models

**José Javier Martín-Gómez** [1], **Agnieszka Rewicz** [2], **Klaudia Goriewa-Duba** [3], **Marian Wiwart** [3], **Ángel Tocino** [4] **and Emilio Cervantes** [1,*]

1  IRNASA-CSIC (Institute of Natural Resources and Agronomy-Consejo Superior de Investigaciones Científicas), Cordel de Merinas, 40, E-37008 Salamanca, Spain
2  Department of Geobotany and Plant Ecology, Faculty of Biology and Environmental Protection, University of Lodz, 12/16 Banacha Str., 90-237 Lodz, Poland
3  Department of Plant Breeding and Seed Production, University of Warmia and Mazury in Olsztyn, 10-724 Olsztyn, Poland
4  Departamento de Matemáticas, Universidad de Salamanca, Plaza de la Merced 1, 37008 Salamanca, Spain
*  Correspondence: emilio.cervantes@irnasa.csic.es; Tel.: +34-923-219-606

**Abstract:** Modern automated and semi-automated methods of shape analysis depart from the coordinates of the points in the outline of a figure and obtain, based on artificial vision algorithms, descriptive parameters (i.e., the length, width, area, and circularity index). These methods omit an important factor: the resemblance of the examined images to a geometric figure. We have described a method based on the comparison of the outline of seed images with geometric figures. The J index is the percentage of similarity between a seed image and a geometric figure used as a model. This allows the description and classification of wheat kernels based on their similarity to geometric models. The figures used are the ellipse and the lens of different major/minor axis ratios. Kernels of different species, subspecies and varieties of wheat adjust to different figures. A relationship is found between their ploidy levels and morphological type. Kernels of diploid einkorn and ancient tetraploid emmer varieties adjust to the lens and have curvature values in their poles superior to modern "bread" varieties. Kernels of modern varieties (hexaploid common wheat) adjust to an ellipse of aspect ratio = 1.6, while varieties of tetraploid durum and Polish wheat and hexaploid spelt adjust to an ellipse of aspect ratio = 2.4.

**Keywords:** geometric curves; J index; kernel; image analysis; morphology; seed; shape; wheat

## 1. Introduction

Bread wheat is a major crop with an annual production likely to reach more than 750.4 million metric tons in 2016–2017 (Foreign Agricultural Service, USDA, 2018). The history of agriculture, and in particular, the recent mechanized techniques for product processing, have resulted in a large extension dedicated to the main agricultural varieties. Thus, about 95% of the wheat produced consists of *Triticum aestivum* L., an hexaploid species usually called "common", "bread" or "soft" wheat [1]. The majority of commercial bread sold throughout the world is the product of varieties of *T. aestivum* ssp. *aestivum*, characterised by naked kernels, which is an advantage for milling in the production of bread and pasta, as well as in the baking industry. In contrast, wild species of wheat, along with the domesticated species einkorn, emmer and spelt, have hulled kernels, and their grains are tightly enclosed between glumes making the threshing difficult [2,3].

In recent years, there has been an increase in "alternative" wheat species due to their potential nutritional value [4]. The most common ancient wheat species commercially available are einkorn

(*Triticum monococcum* L.), emmer (*Triticum dicoccum* [Schrank ex Schübl.] Thell.) and spelt (*Triticum spelta* (L.) Thell.). Spelt is the ancestral species from which *T. aestivum* was derived.

*Triticum aestivum* ssp. *aestivum* is an allohexaploid species. It has six sets of chromosomes, two sets from each of three different species. Hybridisation between *T. monococcum* (einkorn wheat) (genome AA) and *Aegilops speltoides* Tausch (genome BB) resulted in the allotetraploid species *Triticum durum* (Desf.) (genomes BBAA). Two additional sets of chromosomes (genome DD) came from wild goat-grass *Aegilops tauschii* Coss. Durum wheat is used mainly in the production of pasta and couscous. Ancient or relict wheats have been reintroduced recently into the markets due to the increasing interest in their nutrient composition and low gluten content [5].

Other tetraploid species in addition to *T. durum* are, for example, Polish wheat (*Triticum polonicum* (L.) Thell.) and emmer (*T. dicoccum*). The introduction of traits in *T. aestivum* ssp. *aestivum* can be made by means of crosses with emmer wheat, the progenitor of both *T. aestivum* ssp. *aestivum* and *T. durum* [6].

Tetra- and hexaploid species have an increased grain size. The conducted morphological analysis of wheat kernels of different species and varieties has revealed interesting differences [7]. For example, the grain of *T. spelta*, the ancestral wheat from which *T. aestivum* ssp. *aestivum* derived, is thinner and longer than the grain of modern bread wheat varieties. This is due most probably to increased yield in the milling of more spherical grains, and thus selection of this character throughout the years.

Modern methods of shape comparison between two-dimensional figures are based on artificial vision and involve the coordinates of the points in the profile of a figure. With these values, the automated application of a series of algorithms permits to calculate several parameters for each figure such as: its longer and shorter diameters, perimeter, centroid, area and circularity index [8,9]. This allows to perform a shape comparison between two or more figures and the subsequent statistical analysis reveals differences between groups of figures. Overall, artificial vision techniques are broadly applied to biological systems, and, in particular, to the seed shape of many species [10–13], but they present an important drawback. During the analytical process, these methods omit an important factor of the resemblance of seed images to a geometric figure.

The method of seed shape quantification involves a comparison with geometric models. In recent years, we have explored geometric models for shape description and quantification in seed images. The cardioid figure is frequent in biological objects because it represents the result of growth of an organ derived from a fixed point [14]. The cardioid was successfully applied for the description of seed shape first in the model plant *Arabidopsis thaliana* (L.) Heynh [15], then in the model legumes *Lotus japonicus* L. and *Medicago truncatula* Gaertn. [16], later on *Capparis spinosa* L. [17], as well as in species of the Papaveraceae, Caryophyllaceae and Malvaceae [18,19], and other species such as *Rhus tripartita* (Ucria) Grande in the family Anacardiaceae [20]. Seeds in many taxonomical groups adjust better to other models different than the cardioid, such as, for example, the ovoid or the ellipse. Seeds in the Cucurbitaceae adjust to the ovoid [21] and in the Euphorbiaceae to the ellipse, such as, for example, species of genus *Ricinus* and *Jatropha* [22,23]. From the geometric viewpoint, a seed cannot be at the same time cardioid, ovoid, and elliptical, and geometric shapes differ according to the taxonomical group. In the case of wheat, a variation of kernel shape may be a consequence of breeding leading to forms with increased yield. The geometric models described here contribute to the accurate description of the differences between varieties.

Our objective was to describe the shape of kernels of wheat species based on their comparison with geometric figures as well as to validate the results of digital image analysis [7]. The comparison of seed images with adequate models for each species and variety allows for the description and classification of wheat kernels in morphological types. In addition, seed shape quantification may be an important tool to verify the results of artificial vision technologies, as well as in the identification of the molecular basis underlying differences in shape.

## 2. Materials and Methods

### 2.1. Species and Varieties of Wheat

The analysis included six wheat taxa (*T. monococcum*, *T. dicoccum*, *T. durum*, *T. polonicum*, *T. aestivum* ssp. *aestivum*, *T. spelta*), of which *T. durum* is represented by three varieties and *T. aestivum* ssp. *aestivum* by two additional varieties. As a result, we have the following: 1. *T. monococcum* (einkorn; Tm); 2. *T. dicoccum* (emmer; Tdi); 3. *T. durum* (var. Floradur; TduF), 4. *T. durum* (var. Duromax; TduD), 5. *T. durum* (var. Duroflavus; TduDu), 6. *T. polonicum* (Polish wheat; Tp), 7. *T. aestivum* ssp. *aestivum*, (Taa), 8. *T. aestivum* ssp. *aestivum* (var. Zebra; TaaZ), 9. *T. aestivum* ssp. *aestivum* (var. Torka; TaaT), 10. *T. spelta* (Ts). All lines and varieties were reproduced at the Department of Plant Breeding and Seed Production of the University of Warmia and Mazury in Olsztyn, Poland. The lines were obtained by the reproduction of accessions obtained from the National Centre for Plant Genetic Resources (NCPGR), Radzików, Poland, the National Plant Germplasm System (NPGS), USA, and Leibniz Institute of Plant Genetics and Crop Plant Research (IPK) in Gatersleben, Germany. A field experiment was conducted in 2014/2015 in the Agricultural Experiment Station in Bałcyny (53°36′ N, 19°51′ E), Poland. Plots with an area of 9 m$^2$ each were established on soil typically used for wheat cultivation [7].

### 2.2. Images of Wheat Kernels

Images of wheat kernels were of two types: (a) the original scan images obtained previously [7] consisted of 150 kernels for each of six wheat taxa (Tm, Tdi, TduF, Tp, TaaZ and Ts), and (b) the photographs taken with a digital camera Sony ILCE 5100.

### 2.3. General Morphological Description by Image Analysis

Scan images from different subspecies were used to obtain data of the area (A), the perimeter (P), the length of the major axis (L), the length of the minor axis (W), the aspect ratio (AR is the ratio L/W), circularity (C) and roundness (R). All these magnitudes were calculated with Image J program [24]. The circularity Index was calculated as described by [8] and roundness as described by [25–27].

### 2.4. Comparison with Geometric Models

The method used for the morphological description of kernel shape involves a comparison with geometric figures used as models. Three geometric models were used: (1) an ellipse of aspect ratio (AR) = 1.8 was chosen to fit the "round varieties" (*T. aestivum* ssp. *aestivum* var. Zebra and Torka), (2) a lens of AR = 3.2 was chosen to fit the elongated kernels (*T. monococcum*), and (3) an ellipse of AR = 2.4 (Figure 1). For each of these geometric models, the ratios were taken from the mean values in the varieties better adapted (*T. aestivum* ssp. *aestivum*, var. Zebra for the ellipse and *T. monococcum* for the lens). Images containing the kernel photographs and the geometric model were composed with Corel PHOTO-PAINT X7 (Corel corporation, v. 17.5.0.907; www.corel.com).

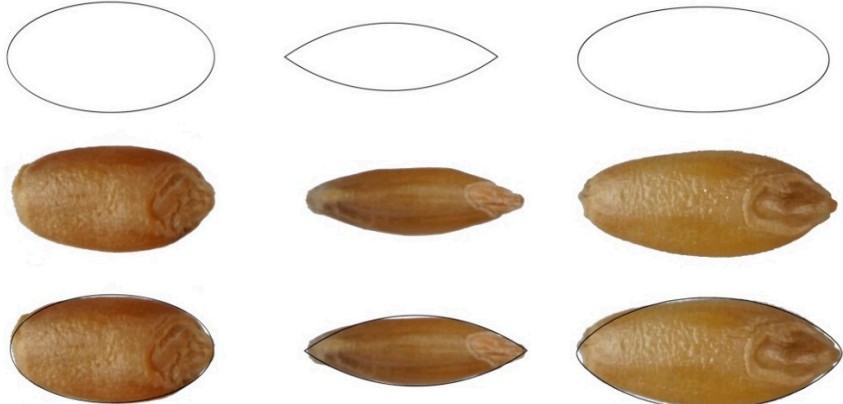

**Figure 1.** Geometric models for the quantification of shape in wheat kernels are: an ellipse of AR (Aspect Ratio) = 1.8 (left), a lens of AR = 3.2 (centre) and an ellipse of AR = 2.4 (right). The kernels in each case correspond to TaaZ (*T. aestivum* ssp. *aestivum* (var. Zebra)), Tm (*T. monococcum* (einkorn)) and TduF (*T. durum* (var. Floradur). Bar equals 1 cm.

Areas of kernel images were calculated with ImageJ program. To obtain the J index, the areas in two regions were compared: the region shared by the model and the seed image (common region, C) and the region not shared between both areas (D). The J index is defined by:

$$J = \frac{(\text{area C})}{(\text{area C} + \text{area D})} \times 100 \tag{1}$$

where C represents the common region and D the regions not shared (Figure 2). Note that J is a measure of seed shape, not of its area. It ranges between 0 and 100 decreasing when the size of the not-shared region grows and equals 100, when the geometric model and the seed image areas coincide, that is, when area (D) is zero. Similarity was considered when J index values were over 90.

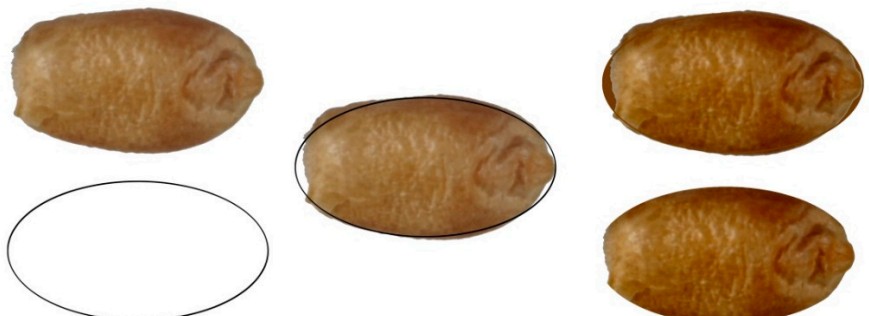

**Figure 2.** Method for obtaining the J index (percentage of similarity between two images, the geometric figure and the kernel image). On the left: the kernel (TaaZ, top) and the geometric model (ellipse, bottom). In the center, the composed image containing both, the kernel and the ellipse. Right: total surface occupied by both figures (top) and shared surface (bottom). The J index is the ratio between the shared and total surface × 100.

*2.5. Curvature Values*

Curvature is a property of smooth curves that indicates the degree of divergence from a straight line by means of measuring the variation of the tangent at each point during progression through the curve. Formally, if $\theta$ represents the angle which the tangent vector makes with the x-axis, the curvature at a point of the curve represents the rate of change of $\theta$ with respect to the arc length s of the curve:

$$K = \frac{d\theta}{ds} \tag{2}$$

To obtain curvature values in the poles of the kernel, first a smooth curve was adjusted to the polygonal obtained from the points selected along the silhouette of the poles. For this purpose, we used Bézier curves [28–30]). Given a representative set of points, an equation based on Bernstein polynomials that reproduces the silhouette of the poles is obtained for each of the two kernel poles (Figure 3).

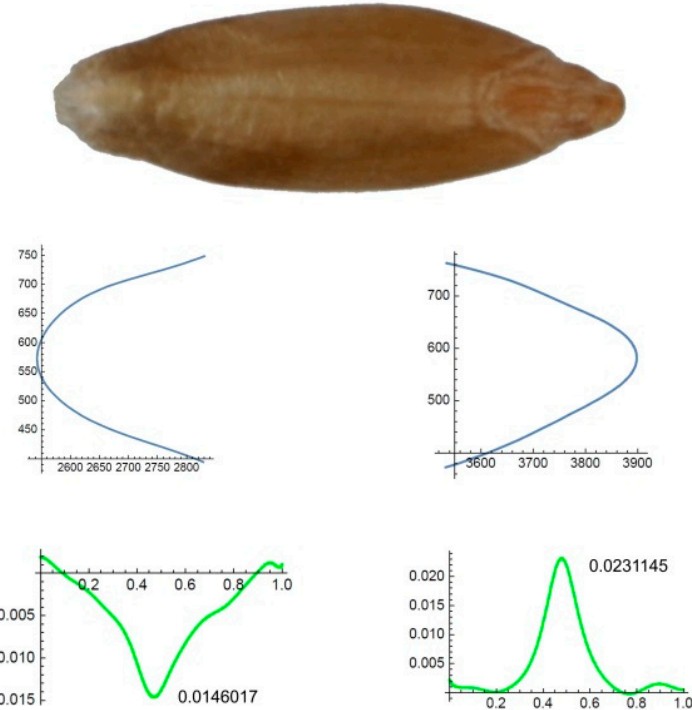

**Figure 3.** Typical kernel of *T. monococcum* (Tm). Graphs of the Bézier equations corresponding to both, basal (left) and apical (right) poles. Graphs representing curvature values in each curve, with an indication of the maximum values for each pole.

### 2.6. Statistical Analysis

ANOVA(Analysis of variance) was used to show significant differences between populations for the measured variables, followed by Scheffé's test to provide specific information on which means were significantly different from one another. The analysis was done with software IBM SPSS statistics v25 (IBM corp., 25.0.0.1, https://www.ibm.com/analytics/spss-statistics-software).

## 3. Results

### 3.1. General Morphological Description

Table 1 shows the mean values of the area (A), the perimeter (P), the length of the major axis (L), the length of the minor axis (W), the aspect ratio (AR is the ratio L/W), circularity (C) and roundness (R) obtained for the kernel images of six taxa of wheat and their varieties. The data were obtained from a new analysis of the original scan images described by [7].

**Table 1.** (a) Mean values of the area (A), perimeter (P), length of the major axis (L), length of the minor axis (W), aspect ratio (AR is the ratio L/W), circularity (C) and roundness (R) in the kernels of six wheat taxa and varieties (*N* = 150). Results obtained with original scan images used by Goriewa-Duba et al. (2018). The mean values marked with the same letter in each column do not differ significantly at *p* < 0.05 (Scheffe's test). (b) Mean values of the area (A), perimeter (P), length of the major axis (L), length of the minor axis (W), aspect ratio (AR is the ratio L/W), circularity (C) and roundness (R) in the kernels of six wheat taxa and varieties. Results obtained with digital photographs of kernels of the types described above. The mean values marked with the same superscript letter in each column do not differ significantly at *p* < 0.05 (Scheffe's test).

| (a) | | | | | | | |
|---|---|---|---|---|---|---|---|
| | **A** | **P** | **L** | **W** | **AR** | **C** | **R** |
| Tm (*T. monococcum* [einkorn ]) | 12.94 [a] | 17.46 [b] | 7.22 [b] | 2.28 [a] | 3.19 [e] | 0.53 [a] | 0.32 [a] |
| Tdi (*T. dicoccum* [emmer] | 17.55 [c] | 19.34 [d] | 7.77 [c] | 2.87 [b] | 2.72 [d] | 0.59 [b] | 0.37 [b] |
| TduF (*T. durum* [var. Floradur]) | 18.60 [d] | 18.28 [c] | 7.28 [b] | 3.24 [d] | 2.26 [b] | 0.70 [d] | 0.44 [d] |
| Tp (*T. polonicum* [Polish wheat]) | 19.42 [e] | 20.02 [e] | 7.87 [c] | 3.14 [c] | 2.52 [c] | 0.61 [c] | 0.40 [c] |
| TaaZ (*T. aestivum* ssp. *aestivum* [var. Zebra]) | 14.38 [b] | 15.38 [a] | 5.81 [a] | 3.13 [c] | 1.87 [a] | 0.76 [e] | 0.54 [e] |
| Ts (*T. spelta*) | 19.27 [d.e] | 19.99 [e] | 7.86 [c] | 3.12 [c] | 2.53 [c] | 0.61 [c] | 0.40 [c] |

| (b) | | | | | | | |
|---|---|---|---|---|---|---|---|
| | **N** | **A** | **P** | **L** | **W** | **AR** | **C** | **R** |
| Tm | 21 | 15.86 [a] | 20.32 [b] | 8.17 [b] | 2.53 [a] | 3.23 [d] | 0.49 [a] | 0.31 [a] |
| Tdi | 28 | 20.43 [b.c] | 20.70 [b] | 7.90 [b] | 2.73 [a] | 2.89 [c] | 0.50 [a.b] | 0.35 [b] |
| TduF | 17 | 22.67 [c.d] | 21.37 [b] | 8.26 [b] | 3.58 [c] | 2.31 [b] | 0.64 [c] | 0.43 [c] |
| Tp | 16 | 27.19 [e] | 21.60 [b] | 7.86 [b] | 3.27 [b] | 2.42 [b] | 0.55 [b] | 0.42 [c] |
| TaaZ | 18 | 19.11 [b] | 18.06 [a] | 6.70 [a] | 3.55 [c] | 1.89 [a] | 0.72 [d] | 0.53 [d] |
| Ts | 18 | 24.44 [d] | 20.84 [b] | 7.24 [a] | 3.02 [b] | 2.41 [b] | 0.50 [a.b] | 0.42 [c] |

The results are similar to those obtained by Goriewa-Duba et al. [7] showing differences between varieties for all the parameters tested. Five groups are obtained for A, P, AR, C and R. Kernels of Tm are smaller, while those of *T. polonicum* are larger (Tp > TduF > Tdi). Results based on new images (Table 1b) gave four groups instead of five for A, AR, C and R and two groups for P. Due to the reduced number of kernels used in this experiment, the results were less discriminant, nevertheless the main differences are maintained in both experiments and are consistent with the results reported [7]. Namely: kernels of Taaz have a lower AR than the other taxa and varieties, while kernels of Tm and Tdi have a higher AR. Kernel images of TaaZ have higher values of circularity and in the case of Tm lower values. Tm and Tdi have lower values of roundness (kernels are more elongated).

### 3.2. Comparison with Geometric Models. Values of J Index

The differences in roundness and circularity values, together with the visual examination of kernel shape, suggested different models for the quantification of seed shape. An ellipse of AR = 1.8 was designed to fit the shape of *T. aestivum* ssp. *aestivum* Zebra (TaaZ) (more rounded), a lens of AR = 3.2 to fit the shape of *T. monococcum* and *T. dicoccum*, and an ellipse of AR = 2.4 to fit the shape of the rest of species, subspecies and varieties (Figure 4).

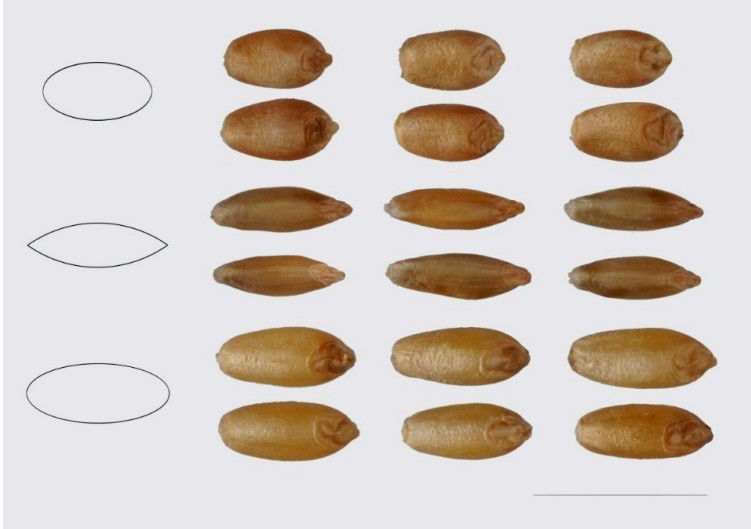

**Figure 4.** Left: The three figures used as geometric models are an ellipse of aspect ratio = 1.8, a lens of AR = 3.2 and an ellipse of aspect ratio = 2.4. When compared to wheat kernels, these give higher J index values respectively with *T. aestivum* ssp. *aestivum* cv. Zebra (two files at the top of the figure), *T. monococcum* (two files in the middle of the figure), and *T. durum* cv. Floradur (two files below). Bar corresponds to 1 cm.

Table 2 shows the statistical parameters and results of comparison of means for values of J index with an ellipse of aspect ratio = 1.8, a lens of aspect ratio = 3.2 and an ellipse of aspect ratio = 2.4 respectively.

**Table 2.** (a) Mean (±SD), minimum and maximum values of J index. Values obtained with an ellipse of AR = 1.8. The mean values marked with the same letter in the superscript do not differ significantly at $p < 0.05$ (Scheffe's test). (b) Mean (±SD), minimum and maximum values of J index values obtained with a lens of AR = 3.2. The mean values marked with the same letter do not differ significantly at $p < 0.05$ (Scheffe's test). (c) Mean (±SD), minimum and maximum values of J index values obtained with an ellipse of AR = 2.4. The mean values marked with the same letter differ not significantly at $p < 0.05$ (Scheffe's test).

| (a) | | | | |
|---|---|---|---|---|
| | N | Mean | Min | Max |
| Tm | 21 | 68.1 [a] (±2.70) | 62.7 | 72.5 |
| Tdi | 28 | 69.7 [a] (±3.57) | 62.3 | 76.5 |
| TduF | 18 | 85.1 [b] (±2.10) | 81.4 | 88.1 |
| TduD (*T. durum* [var. Duromax]) | 13 | 83.8 [b] (±3.69) | 75.8 | 90.4 |
| TduDu (*T. durum* [var. Duroflavus]) | 17 | 83.4 [b] (±3.84) | 73.4 | 88.3 |
| Tp | 16 | 82.6 [b] (±5.26) | 73.0 | 89.7 |
| Taa (*T. aestivum* ssp. *aestivum*) | 16 | 82.0 [b] (±3.04) | 77.5 | 87.8 |
| TaaZ | 18 | 93.5 [c] (±1.32) | 90.7 | 95.4 |
| TaaT (*T. aestivum* ssp. *aestivum* [var.Torka]) | 16 | 93.5 [c] (±1.59) | 89.9 | 96.2 |
| Ts | 18 | 83.6 [b] (±5.78) | 74.6 | 94.2 |

**Table 2.** *Cont.*

| (b) | | | | |
|---|---|---|---|---|
| | **N** | **Mean** | **Min** | **Max** |
| Tm | 21 | 90.4 [e] (±1.84) | 86.8 | 93.3 |
| Tdi | 28 | 90.1 [e] (±2.17) | 84.7 | 93.6 |
| TduF | 18 | 75.6 [b–d] (±4.81) | 66.0 | 84.6 |
| TduD | 13 | 77.1 [c,d] (±3.35) | 72.5 | 83.7 |
| TduDu | 17 | 72.2 [a–c] (±3.92) | 66.2 | 81.2 |
| Tp | 16 | 73.8 [a–d] (±3.37) | 69.6 | 81.6 |
| Taa | 16 | 76.7 [c,d] (±4.13) | 67.4 | 82.9 |
| TaaZ | 18 | 70.3 [a] (±3.73) | 64.9 | 78.3 |
| TaaT | 16 | 71.1 [a,b] (±3.16) | 66.5 | 77.2 |
| Ts | 18 | 77.8 [d] (±4.70) | 70.6 | 86.4 |

| (c) | | | | |
|---|---|---|---|---|
| | **N** | **Mean** | **Min** | **Max** |
| Tm | 21 | 79.4 [a] (±1.84) | 73.7 | 83.4 |
| Tdi | 28 | 85.6 [b] (±2.72) | 80.9 | 91.0 |
| TduF | 18 | 92.8 [c] (±1.59) | 89.3 | 94.7 |
| TduD | 13 | 92.0 [c] (±1.97) | 87.6 | 94.3 |
| TduDu | 17 | 91.9 [c] (±1.76) | 88.8 | 94.8 |
| Tp | 16 | 91.4 [c] (±2.02) | 87.6 | 94.9 |
| Taa | 16 | 92.8 [c] (±1.24) | 90.9 | 95.0 |
| TaaZ | 18 | 80.5 [a] (±3.54) | 75.3 | 87.9 |
| TaaT | 16 | 78.4 [a] (±3.70) | 72.0 | 85.1 |
| Ts | 18 | 91.5 [c] (±2.77) | 86.5 | 94.9 |

The results of the three models show that wheat kernels are classified into three groups according to their shape: (1) a group of "rounded" kernels, (2) a group of "elongated" kernels, and (3) an intermediate group. The first group includes the kernels of TaaT and TaaZ with values of J index over 90 with an ellipse of AR = 1.8. The second group contains Tm and Tdi with values of J index over 90 with a lens of AR = 3.2. The rest of examined species, subspecies and varieties are included in the third group and all of them give values over 90 with an ellipse of AR = 2.4.

### 3.3. Curvature Analysis

Table 3a shows the statistical parameters and results of comparison of means corresponding to the values of curvature obtained for the basal pole of kernels of ten genotypes of wheat.

**Table 3.** (a) Statistical parameters of maximum curvature values in the basal pole of the kernel. The mean values marked with the same letter do not differ significantly at $p < 0.05$ (Scheffe's test). (b) Statistical parameters and results of comparison of means for the ratio between maximum curvature values in the anterior/posterior poles of the kernel. The mean values marked with the same letter do not differ significantly at $p < 0.05$ (Scheffe's test).

| (a) | | | | | |
|---|---|---|---|---|---|
| | N | Mean ($\times 10^{-3}$) | SD | Min ($\times 10^{-3}$) | Max ($\times 10^{-3}$) |
| Tm | 6 | 15.0 [b] | $1.56 \times 10^{-3}$ | 13.0 | 17.7 |
| Tdi | 6 | 24.6 [c] | $6.75 \times 10^{-3}$ | 19.2 | 36.6 |
| TduF | 6 | 6.07 [a] | $1.01 \times 10^{-3}$ | 4.84 | 7.69 |
| TduD | 6 | 8.15 [a] | $8.80 \times 10^{-4}$ | 7.07 | 9.10 |
| TduDu | 6 | 6.32 [a] | $1.43 \times 10^{-3}$ | 5.21 | 9.03 |
| Tp | 6 | 5.82 [a] | $8.10 \times 10^{-4}$ | 5.02 | 7.18 |
| Taa | 6 | 7.01 [a] | $1.34 \times 10^{-3}$ | 5.20 | 9.28 |
| TaaZ | 6 | 6.05 [a] | $9.10 \times 10^{-4}$ | 4.64 | 7.09 |
| TaaT | 6 | 8.44 [a] | $2.08 \times 10^{-3}$ | 6.35 | 12.2 |
| Ts | 6 | 9.59 [a] | $2.70 \times 10^{-3}$ | 6.13 | 12.3 |
| (b) | | | | | |
| | N | Mean | SD | Min | Max |
| Tm | 6 | 1.754 [a] | 0.236 | 1.438 | 2.118 |
| Tdi | 6 | 1.049 [a] | 0.483 | 0.700 | 1.973 |
| TduF | 6 | 1.865 [a] | 0.238 | 1.537 | 2.186 |
| TduD | 6 | 1.777 [a] | 0.588 | 1.218 | 2.515 |
| TduDu | 6 | 1.837 [a] | 0.270 | 1.537 | 2.290 |
| Tp | 6 | 2.905 [b] | 0.419 | 2.226 | 3.364 |
| Taa | 6 | 1.827 [a] | 0.541 | 1.063 | 2.624 |
| TaaZ | 6 | 1.614 [a] | 0.750 | 0.881 | 2.918 |
| TaaT | 6 | 1.153 [a] | 0.521 | 0.530 | 1.768 |
| Ts | 6 | 1.499 [a] | 0.591 | 0.984 | 2.632 |

The two subspecies belonging to group 2 (elongated kernels, Tm and Tdi), giving high values of J index with a lens of AR = 3.2, have higher curvature values in the basal pole (Figure 5). Similar results were obtained with the apical pole (Figure 5). Overall, curvature values were higher in the anterior pole due to a maximum in the apex region of the embryo.

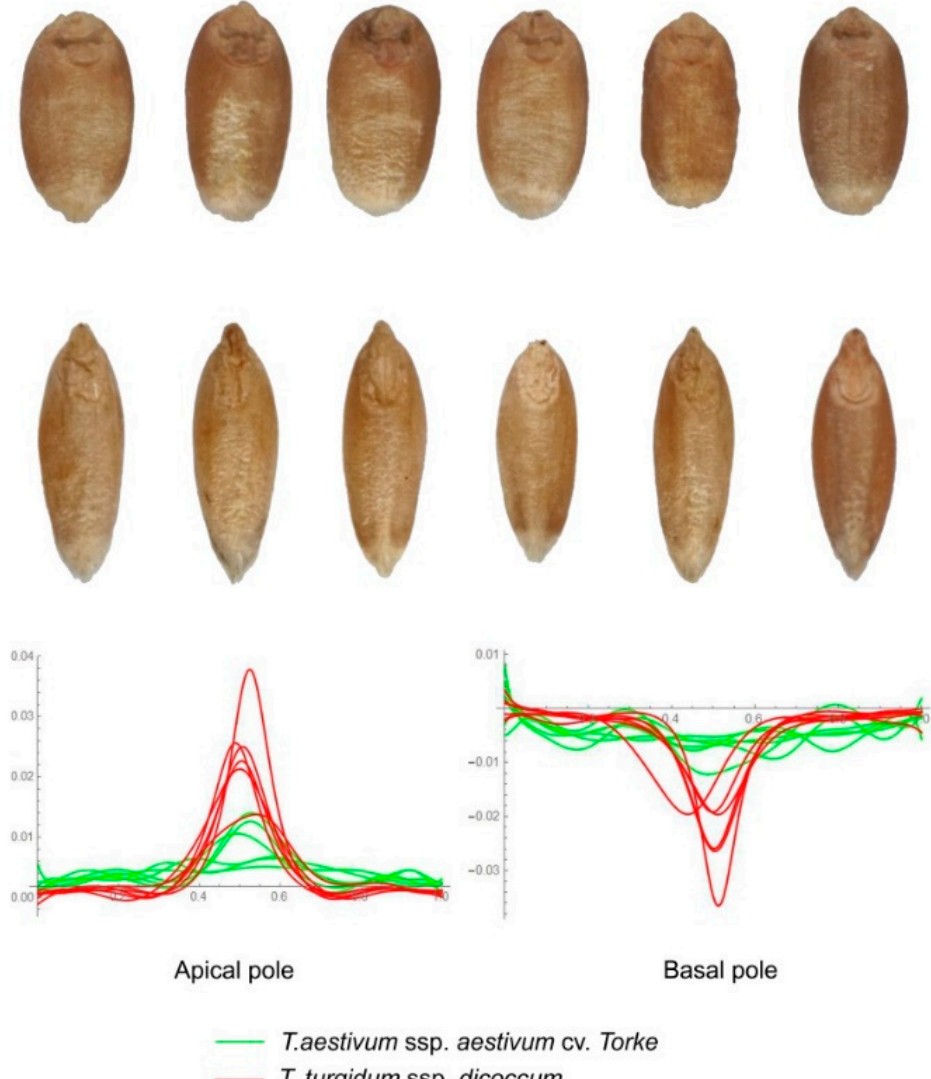

**Figure 5.** Above: typical seeds of *T. aestivum* ssp. *aestivum* Zebra (TaaZ). Centre: typical seeds of *T. monococcum* (Tm). Below, left: Curvature values in the apical pole for TaaZ (green) and Tm (red). Below, right: Curvature values in the basal pole for TaaZ (geen) and Tm (red).

### 3.4. Differences in Symmetry

The ratio between maximum values of curvature in both poles gives an idea of the symmetry of the kernel. Values close to 1 reflect similar curvature in both poles, and thus, high symmetry. The more this ratio diverges from 1, the more asymmetry is displayed by the kernel.

When the comparison analysis included the results for all species and varieties, only *T. polonicum* (Tp) diverged from the others (Table 3b). In Tp, the ratio between maximum curvature values in the anterior and posterior poles is higher than in the other objects, thus representing higher asymmetry in the kernel of this wheat (Figure 6).

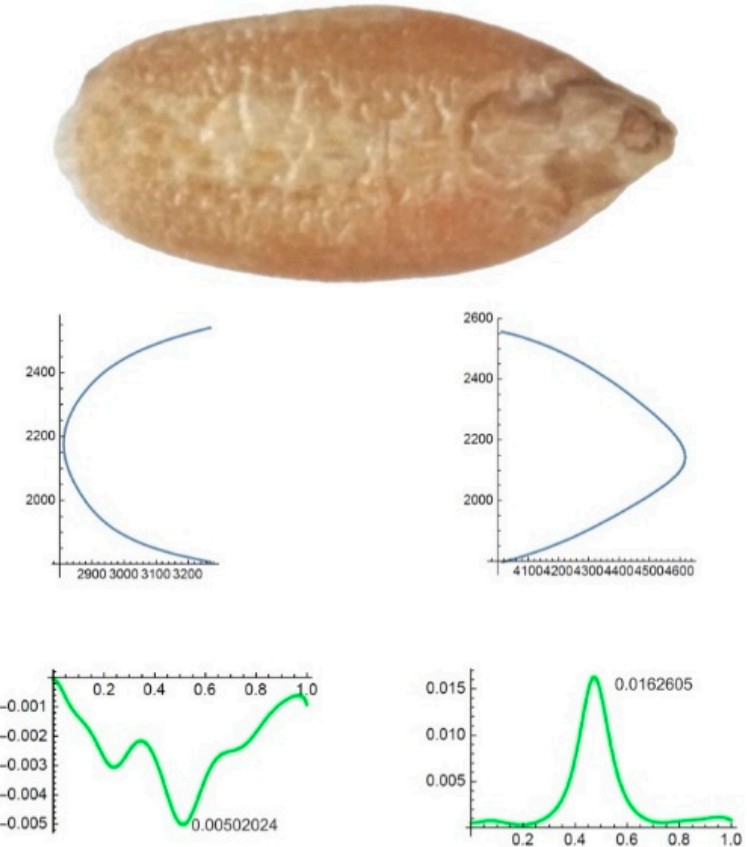

**Figure 6.** Asymmetry in the kernel of *T. polonicum*. Above: typical kernel. Centre: Graphic representation of the poles (Bezier curves). Below: Curvature values.

## 4. Discussion

Automated methods based on artificial vision and algorithms are applied in general to all kinds of shape comparisons in remote sensing and morphology. In botany, these methods have been applied to the analysis and classification of seeds in diverse taxonomic groups [10–13], as well as for wheat kernels [7].

The comparison of kernels belonging to different groups based on automated methods departs from the coordinate values (x, y) of the points in the contour of the object. Algorithms are applied to calculate data of size (the area, perimeter, length and width) as well as shape descriptive variables (the roundness and circularity index). The comparison of different images by statistical analysis often includes together data of size and shape providing a final mixture whose results are highly dependent on size variables while circularity and roundness values only give a relative idea of shape. The groups obtained by this method do not take into account important information concerning the similarity of a seed image to a geometric figure. In contrast, the J index provides a measurement of the similarity to geometric figures, this having two advantages [15,16,21,31]. First, the groups resulting from each experiment are associated with geometric figures. Second, the method provides a control for results coming from different experiments.

The results presented here were obtained in two steps. First, a preliminary general morphological analysis was performed comparing independently data concerning size and shape. Later on, results are concentrated exclusively on shape independently of size. The first part involves semi-automated data analysis, the latter applies the method based on the comparison of kernel images with geometric figures [15,16,31].

Diploid wheat species have a smaller grain than tetraploid wheat [7,32]. This is confirmed in our preliminary study where diploid einkorn was characterized by the smallest kernel area. Tetraploid

species have a larger surface area. The relationship between the ploidy level and size, however, is not maintained in all hexaploid wheats. For example, kernels of ancient spelt are large, while those of the commercial varieties Torka and Zebra are smaller [7]. While size can be dramatically variable due to growing conditions and position of the kernel in the wheat spike the kernel shape may be quite stable.

Concerning shape, three groups result from the comparison of wheat kernels with geometric figures: the ellipse (AR = 1.8), the lens (AR = 3.2) and the ellipse (AR = 2.4). The first comprises the commercial hexaploid varieties TaaZ and TaaT; the second the ancient diploid *T. monococcum* (Tm; einkorn) and the ancient tetraploid *T. dicoccum* (Tdi; emmer), and the third group contains the remaining species, subspecies and varieties, including the three varieties of tetraploid *T. durum* (TduF, TduD, TduDu), *T. polonicum* (Tp) and two hexaploids (Taa and Ts).

In general, a trend is observed with the ancestral taxa being more elongated and less round than the modern wheat bread varieties, which is in line with the selection done in favor of rounded kernels associated with more yield in the milling process [7,33,34]. The figures used present a gradient of roundness that parallels the evolution of wheat kernels from the more primitive diploid einkorn and early tetraploid emmer varieties that resemble an elongated lens of AR = 3.2, through the intermediate shape of modern tetraploid varieties and ancient hexaploid *T. aestivum* ssp. *aestivum* and *T. spelta*, resembling an ellipse of AR = 1.8, to finally more rounded kernels of modern hexaploid varieties TaaZ and TaaT.

The values of curvatures in the apical and basal poles may reveal differences between genotypes in the group composed by a higher number of species, subspecies and varieties. The kernels of *T. polonicum* present a particular asymmetry that may represent a differential characteristic this species. It may be interesting to investigate those aspects in other wheat species and varieties.

The method reported here of shape comparison with geometric figures is useful for the visualization and validation of results obtained by more commonly used automated artificial vision methods. The groups obtained by digital image analysis correspond to kernel types resembling different geometric figures and the similarity to figures can be quantified for each group by J index. Similarly this method may be of practical application in the validation of results of digital analysis applied to shape comparison in the identification of varieties, species description and in general, in taxonomy.

## 5. Conclusions

(1) The comparison of kernel images with geometric figures allows a classification of kernels based on their similarity to the figures of the lens and different ellipses. (2) Kernels of diploid einkorn and ancient tetraploid emmer varieties adjust to the lens. Kernels of modern varieties (hexaploid common wheat) adjust to an ellipse of Aspect Ratio = 1.6, while varieties of tetraploid durum, emmer and Polish wheat and hexaploid spelt adjust to an ellipse of Aspect Ratio = 2.4. (3) Adjust to a lens is associated with higher curvature values in the poles of ancient varieties. (4) The kernels of *T. polonicum* present a particular asymmetry that may represent a differential characteristic of this species. (5) The method may be of practical application in the validation of results of digital analysis applied to shape comparisons.

**Author Contributions:** Conceptualization, Á.T. and E.C.; Data curation, J.J.M.-G.; Formal analysis, J.J.M.-G., Á.T. and E.C.; Investigation, J.J.M.-G., A.R., K.G.-D., M.W., Á.T. and E.C.; Methodology, J.J.M.-G., Á.T. and E.C.; Software, J.J.M.-G. and Á.T.; Supervision, E.C.; Validation, J.J.M.-G., A.R., K.G.-D., M.W., Á.T. and E.C.; Writing—original draft, E.C.; Writing—review & editing, J.J.M.-G., A.R., K.G.-D., M.W., Á.T. and E.C.

**Funding:** This research was funded by the National Science Centre, Poland (project Preludium 14 [2017/27/N/NZ9/00058]) and Universidad de Salamanca (Programa XIII para la financiación de grupos GIR).

**Conflicts of Interest:** The authors declare no conflict of interest.

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
