# Peer review of "Morphological Description and Classification of Wheat Kernels Based on Geometric Models"

_agronomy, doi:10.3390/agronomy9070399_

Round 1

Reviewer 1 Report

1. The paper is overloaded by tables, many of which can be merged (e.g., 1 & 2, 3-5, 6 & 7) or even omitted.

2. There are many formatting errors, which I indicated and that should be corrected. Please see attached file

3. In general paper represents continuation of original conceptual and mostly theoretical studies of Martín-Gómez & Cervantes extrapolated on cereals. However, just like in previous articles of these authors, it is not clear how these investigations can be applied and what they really provide for us. The main conclusion of current paper, as I understand, is that modern varieties of Triticum have lower AR index than ancestral. But so what? It is not necessary to apply any special mathematic algorithm to see that difference, which is more or less obvious. So, I guess, it would be great if authors will provide us with more clear argumentations and explanation of usefulness of their research.

Author Response

Dear Reviewer ,

Thank you very much for your comments that have contributed to improve the quality of the article. These have been taken into account for the new version.

The tables have been merged (1 & 2, 3-5, 6 & 7).

All the formatting errors indicated in the attached PDF file have been corrected. In the case that many references are quoted simultaneously in the text (under the same parenthesis), according to the instructions to authors there is no space between references. For example we write [1,2,3] instead of [1, 2, 3].

Concerning point 3: the practical usefulness of this work, this is an important aspect that both evaluators have remarked and has now been highlighted by a modification at the end of the introduction, a complete new paragraph at the end of the discussion and an additional conclusion. The last paragraph in the introduction states now:

"Our objective was to describe the shape of kernels of wheat species based on their comparison with geometric figures as well as to validate the results of digital image analysis [7]. The comparison of seed images with adequate models for each species and variety allows for the description and classification of wheat kernels in morphological types. In addition, seed shape quantification may be an important tool to verify the results of artificial vision technologies as well as in the identification of the molecular basis underlying differences in shape."

And the last paragraph of the discussion says:

The method reported here of shape comparison with geometric figures is useful for the visualization and validation of results obtained by more commonly used automated artificial vision methods. The groups obtained by digital image analysis correspond to kernel types resembling different geometric figures and the similarity to figures can be quantified for each group by J index. Similarly this method may be of practical application in the validation of results of digital analysis applied to shape comparison in the identification of varieties, species description and in general, in taxonomy.

In addition conclusion 5 has been added in the same direction:

5) The method may be of practical application in the validation of results of digital analysis applied to shape comparisons.

Reviewer 2 Report

I found the article very well written. I only spotted one typo, in which “than” was spelled as “that” in line 186.

My biggest concern is the practical usefulness of this work. Is this a pure science, simply for fun, or is it to be useful economically?  In practice, the kernel shape may be quite stable but the size can be dramatically variable due to growing conditions and position of the kernel in the wheat spike. Among my personal experience in a what genotype which can produce up to 6 kernels within a single spikelet. The fist seeds can be 5 times larger than the smallest. Does that mean 6 models are needed to describe a single wheat cultivar, even in a single growing environments. This question must be addressed in the interaction as well as in the discussion.

Author Response

Dear Reviewer,

Thank you very much for your comments that have contributed to improve the quality of the article. These have been taken into account for the new version.

On line 186 that has been changed to than.

Concerning the relationship between size and shape, the work has been done without a selection of kernels by size and thus variation in size can be important. Nevertheless, we agree with you when you say that while size can be dramatically variable due to growing conditions and position of the kernel in the wheat spike, the kernel shape may be quite stable. This important aspect has been now introduced in the discussion (end of 4th paragraph).

Concerning the practical usefulness of this work, this is an important aspect that both evaluators have remarked and has now been highlighted by a by a modification in the end of the introduction, a complete new paragraph at the end of the discussion, and a new conclusion (number 5).

The last paragraph of the introduction indicates now:

Our objective was to describe the shape of kernels of wheat species based on their comparison with geometric figures as well as to validate the results of digital image analysis [7]. The comparison of seed images with adequate models for each species and variety allows for the description and classification of wheat kernels in morphological types. In addition, seed shape quantification may be an important tool to verify the results of artificial vision technologies as well as in the identification of the molecular basis underlying differences in shape.

The last paragraph of the discussion has been added and says:

The method reported here of shape comparison with geometric figures is useful for the visualization and validation of results obtained by more commonly used automated artificial vision methods. The groups obtained by digital image analysis correspond to kernel types resembling different geometric figures and the similarity to figures can be quantified for each group by J index. Similarly this method may be of practical application in the validation of results of digital analysis applied to shape comparison in the identification of varieties, species description and in general, in taxonomy.

Conclusion 5 has been added:

5) The method may be of practical application in the validation of results of digital analysis applied to shape comparisons.